# Perception of E-cigarette Use among Adult Users in China: A Mixed-method Study

**DOI:** 10.3390/ijerph17082754

**Published:** 2020-04-16

**Authors:** Duan Zhao, Yantao Zuo, Dilshat S. Urmi, Yangyujin Liu, Pinpin Zheng, Wang Fan, Abu S. Abdullah

**Affiliations:** 1Global Health Research Center, Duke Kunshan University, Suzhou 215347, China; dz62@duke.edu (D.Z.); dilshaturmi@yahoo.com (D.S.U.); yangyujin.liu@dukekunshan.edu.cn (Y.L.); 2Rutgers Cancer Institute of New Jersey, New Brunswick, NJ 08903, USA; 3Department of Preventive Medicine, School of Public Health, Fudan University, Shanghai 20032, China; zpinpin@shmu.edu.cn; 4Department of Politics, East China Normal University, Shanghai 200241, China; wangfan512@126.com; 5Duke Global Health Institute, Duke University, Durham, NC 27710, USA; 6Boston University School of Medicine, Boston Medical Center, Boston, MA 02118, USA

**Keywords:** electronic cigarette, IQOS, smoking, perception, tobacco harm reduction, Chinese

## Abstract

The use of electronic cigarettes (ECs) is increasing rapidly in China, but the perception of these products and their health impact among Chinese users have received little research attention. This study collected semi-structured in-depth interview data from experienced adult EC (including heated tobacco products also named ECs on the Chinese market) users in the Shanghai area. The subjects were recruited from those who participated in a previous online survey on EC use. A thematic narrative approach was used to analyze the data. Thirty current EC users were interviewed for evaluation of their perception of EC use in a variety of categories, including reasons for using, concerns, social acceptance, satisfaction, and health impacts. Participants’ common reasons for using ECs were the ease of use and carrying, hygiene, no fire hazard, reduced smoke exposure, aiding quitting smoking, reduced health hazard, palatable flavor, being fashionable, and substituting cigarettes in non-smoking areas. Most (90%; 27/30) participants reduced (77%) or quit smoking (13%) after using ECs, and 80% were willing to recommend these products to others. Most (90%) of the participants reportedly noticed positive health changes after using ECs. Regulatory concerns were expressed by 33% participants. Participants predominately viewed ECs as a viable substitute for smoking, with substantial effects on tobacco harm reduction. These findings lend support to EC use as a promising opportunity for public health promotion in China through engaging smokers in smoking cessation attempts. However, overall public health benefits/risks of EC use, and its regulatory affairs need to be considered.

## 1. Introduction

Electronic cigarettes (ECs), which are hand-held electronic devices that produce inhalable vapor from liquid (namely “e-liquid”) usually containing nicotine, have grown rapidly among adults as a substitute for traditional cigarettes around the world [1]. Meanwhile, heated tobacco products (HTPs; e.g., IQOS—“I quit original smoking”) have recently been reintroduced to the mass market. While HTPs are classified as tobacco products in some countries such as the U.S., they are marketed as a variation of ECs in other countries such as Korea [2]. In China, both HTPs and heated liquid ECs are referred to as “dian zi yan’’ (i.e., electronic cigarettes) by sellers and consumers. Unlike other nicotine products, ECs (including HTPs) have largely been sold as mass-marketed consumer products, and their market share has increased sharply [3]. Yet, our knowledge of EC use in China is still limited.

According to a 2015 China Adult Tobacco Survey, 3.1% of people (15+ years old) had tried ECs, and 0.5% were using them [4]. Compared with people in the U.S., European Union, and other countries, Chinese in general knew less about ECs [5]. However, in recent years, EC use is becoming increasingly popular in China, and the domestic sales have also surged [6]. To understand the driving forces behind the current and future use of ECs and their impact on smoking and public health in China, it is important to explore and investigate perceptions of EC use among its users.

Studies of other populations have found that adults have various motivations for EC use, including reduced health risks, replacement of traditional cigarettes with ECs, aiding smoking cessation, and the intake of nicotine in areas where smoking is prohibited [7,8]. These positive attributes are also typical themes in EC advertising and promotion [9]. While some studies have indicated that ECs are generally less harmful than traditional cigarettes [10,11] and switching to ECs can decrease health risks [12], evidence has also shown that EC use may lead to both short-term and long-term health risks [13]. Furthermore, there is still a lack of consensus in the literature on the effectiveness of EC use for smoking cessation [13]. Given that China has the largest tobacco use epidemic in the world with over 40% of adult males smoking cigarettes [14], it is of interest to evaluate the motivational factors underlying current EC use among smokers in China and their concern about the potential risks of EC use.

Although EC use is on the rise in China, the governance and regulation of EC use, and marketing are still lagging behind compared to many other countries [15]. It was argued that the lack of regulation encouraged the use of ECs and its misconceptions [15] among the Chinese. The recent major government intervention banned EC selling online, though its impacts have not been evaluated. Understanding EC users’ perception can inform continued research on the benefits and risks of ECs for reducing tobacco harm and facilitate judicial policy-making actions. This study aimed to assess perceptions of the reasons for use, harmfulness, addictiveness, safety concerns, smoking, and the health impact of ECs among Chinese users.

## 2. Methods

This study used a qualitative design to assess EC use and its perceptions among Chinese adults. Interviews were conducted to inquire about their smoking history, EC use behavior, and perceptions. Trained research assistants (graduate students in the Masters of Global Health program at Duke Kunshan University) conducted 30 semi-structured in-depth interviews (IDI) with EC users in the Shanghai metropolitan area during March 2019. This study was approved by the Institutional Review Board of Duke Kunshan University (Number: 2018ABU026). Written informed consent was obtained from all participants. Each interview lasted for about 50 min. All answers were written down, but not taped, and transcribed into electronic records for analysis afterward.

### 2.1. Participant Eligibility and Recruitment

We previously conducted an online survey on EC use with adult EC users in China. In that survey, EC users were asked to indicate if they were willing to participate in an in-depth interview to talk about their experiences with EC use. Interested individuals (*n* = 30; 2.9% of online survey completers) who lived near Shanghai were reached by telephone to confirm their willingness to participate in a face-to-face interview. Participants were recruited if they were ≥18 years old, used ECs for >6 months, and spoke Mandarin. During the call, the research staff briefly introduced the study, including information about who we are, what the research was about, why we wanted to talk in person, what we did to ensure the confidentiality and autonomy of the gathered information, and the predetermined time schedule for the interviews. At this stage, an initial oral consent to participate in the study was obtained from each eligible participant. Interested participants were invited individually to a quiet coffee store in town. Written informed consent was obtained from all the participants before the interview. Each interview lasted for about 50 min. All answers were written down by both the interviewer and a note-taker on the side, but not taped, and transcribed into electronic records for analysis afterward. Each participant received monetary compensation (200 Chinese Yuan; US$28) for their transportation and time. All the interviews were conducted by a graduate student of Duke Kunshan University (DZ), in Mandarin Chinese, who received extensive training on qualitative data collection and the ethical conduct of research.

### 2.2. Interview Guides

The PIs (Y.Z. and A.S.A.) drafted the first version of the IDI guideline. The guideline was then reviewed by the research team and went through the translation and back-translation process multiple times to avoid misunderstanding and ensure the most appropriate and culturally applicable translation. A total of 39 interview questions were divided into five thematic sections: (1) smoking history and current smoking status (e.g., years of smoking and cigarettes smoked per day); (2) EC use history and current products in use and use patterns (e.g., frequency of use; typical use settings); (3) perceptions of ECs (e.g., addictiveness); (4) perceived impact of EC use on cigarette smoking and health; and (5) participant demographics (e.g., gender, age, and education). Close- and open-ended questions were mixed in the interview in order to elicit both focused and unrestricted answers. The interviewer frequently interacted with participants during the interview process to discuss emerging themes.

### 2.3. Data Analysis

Basic demographic, smoking, EC use status, and quantified EC use perceptions were summarized using descriptive statistics. Microsoft Excel 2016 was used to code transcripts of the interviews. All transcripts were coded by D.Z. and checked by the PI (A.S.A.) using a thematic narrative approach [16]. The analyses were guided by the interview questions, which were developed based on the initial research aims, but emergent themes were also identified. Therefore, our approach combined an inductive and deductive approach. Codes were then compared to identify common EC use experiences and views, as well as to detect cross-cutting themes following the approaches used by the team in other studies in Chinese participants [17]. Any new themes that emerged were explored across all the interview transcripts. Representative quotes for each theme were selected based on their universality across interviews.

## 3. Results

### 3.1. Participant Demographics

Table 1 provides an overview of the sample characteristics. In total, we conducted in-depth interviews with 30 current EC users. The average age of the participants was 35.5 years. Most of our participants were currently using traditional cigarettes (80%), had a bachelor’s or above degree (90%), and were employed (96.7%). Those who were also smokers had been smoking for an average of 13.0 years and currently smoked 7.6 cigarettes on average per day.

### 3.2. EC Use Patterns

As shown in Table 2, most participants (66.7%) started using ECs because of a recommendation from an EC user. The common types of ECs used by the participants were heated liquid products (46.7%) and IQOS (53.3%), a heat-not-burn tobacco product. Twenty-six participants (86.7%) used ECs every day, and most used ECs at irregular times. Most participants (73.3%) had used ECs in non-smoking places. The most common venues included home, private car, workplace or workplace resting area, and restaurants. Some also used ECs in bars, KTVs (karaoke televisions), and public areas.
“I use ECs in all kinds of places, except when I am with a client and we have to smoke traditional cigarettes together.”(Male, 36 years old, IQOS user).

There were gender differences in the perceived patterns of EC use. Female participants commonly used ECs for social purposes. For example, one participant mentioned, “(female) friends would get together at a KTV and vape IQOS.” Unlike male users who often used ECs in workplaces, female users were more likely to only use ECs in private spaces such as homes and personal cars. Compared with male users, female users emphasized more ECs’ fashionable appearance and design. Because of the limited sample size and sampling method, we were unable to draw associations between age and e-cigarette use in this study. However, nine participants mentioned that it was easier for the young to accept ECs as ECs were more fashionable and suitable for new smokers.
“I would not recommend ECs to the elderly any more. They refused my recommendation because they did want to change their behaviors.”(Male, 35 years old, e-liquid EC user).

### 3.3. Perceptions of Using ECs

#### 3.3.1. Motivational Factors for Using ECs

Participants reported diverse reasons for using ECs, including easy to use and carry (no lighter needed), no ashes, reducing cigarette smoking, can help to quit smoking, have a similar taste to traditional cigarettes, but less damage to health, flavorful, fashionable, safe (no fire hazard), can be used in non-smoking places, and do not disturb people around them. Three participants explicitly said the reason for using ECs was to help quit smoking. Almost all participants (29/30) considered health aspects when using ECs. Only one participant used ECs partially for social purposes.
“It’s convenient to smoke ECs. My wife does not allow me to smoke traditional cigarettes.”(Male, 32 years old, e-liquid EC user).
“I don’t cough or feel uncomfortable or have phlegm (when using it). E-cigarettes are convenient, I can use them indoors and do not bother others.”(Male, 40 years old, e-liquid EC user).

However, fifty-seven percent of participants stated that the intensities of the sensation and stimulation of EC vapor were usually not as strong as smoke from traditional cigarettes, though the overall satisfaction rates were similar. Two participants reportedly would turn to traditional cigarettes when their craving for nicotine was strong.

#### 3.3.2. Addictiveness of ECs

Most participants (66.7%) reported that they were a little addicted to ECs, but not as severely as they were addicted to traditional cigarettes. They viewed ECs as “alternatives to traditional cigarettes” and believed that “ECs contain nicotine; thus, addiction is inevitable.” A few others viewed ECs like “soda” or “snacks” that helped them to relax, but they were not indispensable.
“Vaping ECs is like drinking soda. I am not addicted to ECs. When I am craving for nicotine, I will smoke traditional cigarettes.”(Male, 34 years old, e-liquid EC user).

#### 3.3.3. Perceived Health Impact of ECs

Participants generally described ECs as less harmful because they believed that ECs had little or no tar or lower nicotine content as compared with traditional cigarettes. Some articulated that the ECs should have less negative health impact as ECs had a non-irritating taste and that the ECs did not irritate the respiratory tract. However, some IQOS users raised concerns over the tar in the aerosol, while users of e-liquid ECs worried about nicotine and other additives in the e-liquid that might be harmful in the long run.
“I am sure they are harmful because e-cigarettes have nicotine. But they are less harmful (than traditional cigarettes).”(Male, 53 years old, e-liquid EC user).

Four participants (13%) saw television or Internet news reports that said ECs were harmful, thus raising doubts about ECs’ safety. Most participants were not concerned about second-hand EC vapor exposure saying that it was modest, did not irritate people around them, and would not affect other people’s health. Some heated liquid EC users said that there were only water aerosol particles in the vapor from ECs that contained no toxicants. Others argued that nicotine could be harmful as well. Some IQOS users said the aerosol could be harmful to people around, especially for children and pregnant women, as there was still some tar in the aerosol. However, overall, every participant agreed that while ECs were still harmful, the harm was much lesser compared with that of traditional cigarettes. One EC user expressed this by saying:
“It is less harmful to other people. The vapor is not as strong as smoke from traditional cigarettes.”(Male, 22 years old, IQOS user).

Most (90%) of the participants reportedly noticed positive health changes after using ECs. They felt their throat and lung function were getting better, that they did not cough anymore, and had less phlegm.
“ECs are new products. I am not sure whether it is less harmful than traditional cigarettes. But at least I have less phlegm.”(Male, 33 years old, e-liquid EC user).
“I saw the news that said ECs were more harmful than traditional cigarettes.”(Male, 33 years old, IQOS user).

#### 3.3.4. Satisfaction with ECs

Besides the aforementioned perceived intensity difference between ECs and traditional cigarettes, participants were generally satisfied with other aspects of ECs. More than half of the participants argued that the advantages of ECs lied in a wide variety of tastes and/or convenience. IQOS users often complained that IQOS needed to be charged for 7–10 min after one unit was completely vaped so they could not vape more than one IQOS continuously.
“Traditional cigarettes are stronger. But e-cigarettes are safer. At one time I almost scalded my daughter’s head (with a burning traditional cigarette).”(Male, 3 years old, e-liquid EC user).

Participants’ favorite EC flavors were different types of fruit (mentioned 13 times), mint (12 times), original tobacco (8 times), and butter (2 times). Participants liked fruit flavors mainly because they were refreshing, inodorous or even fragrant, and tasty; those who liked mint flavor were mainly attracted by its refreshing taste and being non-irritating to the throat; those who liked the original tobacco flavor reported that the taste of the e-liquid was close to that of the traditional cigarette, making it easier for them to switch from traditional cigarettes to ECs.
“I like watermelon and mango flavors because there is no unpleasant odor in my mouth. Other people are not against me using EC because it smells good.”(Female, 28 years old, e-liquid EC user).

#### 3.3.5. Regulatory Concerns about ECs

One-third of participants were concerned about the regulatory aspects of ECs as there are little or no regulations on the production and sale of ECs in Mainland China. All participants expressed skepticism about the safety and quality control of ECs because they did not trust manufacturers’ promotional claims about their products. Three participants said they received EC related information from public media concerning EC quality and safety. Most participants thought it would be useful to regulate the ECs in China. When asked about health warnings on EC packages, 16 participants replied that they had never noticed them, and only 7 (23.3%) participants had noticed them frequently. Most (79%) heated liquid EC users reported they rarely or never noticed health warnings on the package. In contrast, this number was 50% for IQOS users, despite that warnings on IQOS were reported to be only printed in Japanese.
“I am afraid that the government would limit the use of e-cigarettes which would make their prices rise. Some countries have already banned e-cigarettes.”(Male, 34 years old, e-liquid EC user).

#### 3.3.6. Promotion of ECs to Others

Most participants said they would recommend ECs to other smokers who have not tried ECs. Reasons given by the participants included: ECs are convenient, safe, environmentally friendly, healthier, trendy, and have favorable tastes. However, some argued they would not recommend ECs to those who are not very addicted to nicotine and elders, largely because “the elderly might not be willing to accept these new products.”
“Yes, I will (recommend to others), it is less harmful to me and other people. There will not be phlegm.”(Male, 33 years old, IQOS user).

#### 3.3.7. Prediction of the Trend of EC Use in China

Most participants agreed that ECs have higher acceptability in youths compared with older populations, and ECs have a certain appeal to women due to the presence of many flavors and a stylish appearance. Some participants speculated that older smokers might be too severely addicted to traditional cigarettes for them to switch to ECs. Participants believed that the use of ECs would continue to rise as they had seen more and more people around them starting to use ECs. One participant summarized the trend: “people today pay more attention to health and the expansion of online shopping has helped the spread of EC use.” However, some participants were still skeptical, saying that the government’s actions are still uncertain regarding EC production, use, and regulation in China.

#### 3.3.8. Impact of EC Use on Participants’ Smoking Behavior

Participants commonly viewed ECs as substitutes to cigarettes, which could gradually replace cigarettes altogether. They agreed that the lesser the addiction to nicotine was, the easier it was to switch to ECs. However, others were skeptical, saying that it could be hard to switch as the tastes are different. One important aspect that EC users miss is the social function of traditional cigarettes, and this might be the reason for some EC users to use both EC and traditional cigarettes. One participant mentioned that usually, people give each other cigarettes at business events or in social interactions, which could be hard to refuse.

Of all the participants, 76.7% used fewer traditional cigarettes after starting to use ECs, and 13.3% reportedly had quit smoking traditional cigarettes. For those who still smoked traditional cigarettes, they often would only use ECs at home for family member’s health concerns.
“I am smoking less now. I only smoke traditional cigarettes while I am with my colleagues.”(Male, 38 years old, e-liquid EC user).

#### 3.3.9. The Societal Perception of EC Use

Most participants stated that their peers, family members, and colleagues were usually supportive of their use of ECs, although few participants mentioned that the elderly in the family were sometimes against using ECs because of their lack of understanding of these products. The main reasons for the approval from other people were that smoke exposure and the damage to health are modest and EC use is clean (no ashes). Those who were against ECs were usually non-smokers, and the main reason for opposition was that they believed that ECs still produced secondhand smoke/vapor exposure that is harmful to health.
“My wife supports me to use ECs because my health condition has got better. My friends are against me to use ECs because we cannot share and smoke traditional cigarettes like before.”(Male, 46 years old, IQOS user).

## 4. Discussion

To our knowledge, this is the first in-depth interview study assessing perceptions of EC use among adult users in Mainland China. Findings from these interviews may help us understand how Chinese users think about different aspects of EC use including addictiveness, benefits, and potential harms.

Most of the participants were at least partially satisfied with ECs, but still perceived them as less strong than traditional cigarettes. Similar to previous findings, they also reported reductions in traditional cigarette use after started using ECs [18]. Furthermore, participants claimed they were less addicted to ECs as compared with traditional cigarettes. Their perceptions were consistent with previous findings that ECs have less perceived addictiveness [19,20]. Consistent with other reports [18,21], none of our participants were willing to quit ECs. While this may highlight their addictiveness to nicotine, as many were dual users (concurrently using both ECs and traditional cigarettes), it also underscored their positive attitudes towards ECs.

Consistent with previous research [22], participants in this study reported that they used ECs as a substitution (as least partially) for traditional cigarettes and that ECs helped them reduce smoking. In an earlier study, 20.5% of Malaysian EC users had quit smoking traditional cigarettes over a six-month period since the start of EC use [23]. However, it is unknown whether this reduction of cigarette smoking could lead to long-term smoking abstinence [24,25]. Although the efficacy of ECs to help smokers quit was similar to nicotine patches [26], the confidence in existing clinical trial data was rated “low” by a Cochrane review [27]. Despite insufficient clinical trial evidence for its efficacy, the present results suggested that EC use was perceived by most participants as a promising aid to quitting smoking.

For some participants in this study, one of the main obstacles of quitting smoking was the need for smoking traditional cigarettes in social situations where sharing cigarettes remains a social norm and an expected etiquette among smokers in China [28]. We believe the pressure to conform to the smoking-related social norm was one reason for many of our participants (80%) to remain dual users. Therefore, interventions that promote ECs for smoking cessation should consider this socio-cultural aspect of smoking in China.

Our findings also revealed a distinct perception of EC use between younger and older adults. Most of our participants agreed that ECs are more acceptable to young adults because they are fashionable and young smokers are generally less addicted to nicotine. This is consistent with a previous observation that young adults viewed ECs as novel toys in the era of technology [29]. We also found that male and female users expressed different motivations for using ECs. Similar to previous findings [30], ECs were often viewed as fashionable and sociable by female users in this study. This might be linked to the common perception that using ECs is less stigmatizing for women than smoking traditional cigarettes [31]. Although many of our male participants were also attracted by ECs’ fashionable look, they often chose ECs primarily for health reasons and to bypass no-smoking restrictions in workplaces or public venues. Their results suggested that age- and gender-related individual differences should be considered in prevention and intervention efforts targeting EC use in China.

Although some participants had concerns about the current lack of regulation on ECs’ production and use in China, participants in this study generally believed that ECs were less harmful than traditional cigarettes, consistent with previous findings [32]. Whereas ECs were found to be generally safer than tradition cigarettes [33], public health benefits can only be achieved if EC users do not switch to or concurrently use traditional cigarettes [34]. Unfortunately, there is little evidence that most of the participants who were dual users were aware of the latter information.

We also found that many participants in our study received inaccurate information about the addictiveness, harm, and health effects of ECs. EC users in Australia reported they also had difficulty sourcing practical and trustworthy information [35]. In particular, some participants in our study thought that IQOS did not contain tar or nicotine and heated liquid ECs contained lower nicotine content than traditional cigarettes. However, as a heat-not-burn tobacco product, IQOS does deliver tar, which may contain as much as 20% of the carcinogenic nitrosamine as that of traditional cigarettes, and it remains debatable whether, overall, IQOS is less harmful than traditional cigarettes [34,36]. E-liquid ECs can also produce systemic nicotine levels similar to or even higher than those produced from traditional cigarettes [37]. We also found some that participants received EC-related information from news reported by official media. However, most of such information was related to the production quality of ECs rather than their health impact. Therefore, the government and health educators should provide up-to-date health knowledge about ECs to the public, such as the content of EC vapor and the adverse health impacts of nicotine and toxicants in EC aerosol, as well as the addictiveness and environmental impacts of ECs.

In addition, most of the participants did not notice health warnings on the EC packages as they were often unnoticeable or simply nonexistent. Warning labels on ECs should be noticeable as they may influence perceptions about the risks of these products [38]. Higher risk perceptions of ECs could deter smokers from using ECs as a cessation tool, thus preventing a potential public health benefit, and vice versa [39]. Thus, science-based health warnings on EC packaging should be under future regulations in China.

## 5. Limitations

First, the participants in this study were self-selected individuals who volunteered to participate in the interviews. Therefore, they were likely more enthusiastic about ECs than the average users in China. That the current sample was composed of either past or current traditional cigarette users might limit the results’ generalizability to young non-smokers who try ECs first. Furthermore, all participants were from the Shanghai metropolitan area; thus, the current findings may not be generalizable to EC users in other regions or rural areas in China. In addition, this study can only provide a snapshot of Chinese EC users’ perception at this point as EC use in this country is an ongoing and fast-changing phenomenon. Despise these limitations, the present in-depth interview study, which was among the first of its kind conducted in China, enabled us to gain an understanding of a diverse group of Chinese EC users’ experiences, motivations, and perceptions.

## 6. Conclusions

ECs were considered less harmful than traditional cigarettes and perceived as helpful for reducing or quitting smoking by most Chinese EC user participants in this study. However, many of the participants did not have adequate information about the safety, addictiveness, and harmfulness of ECs. Hence, there is a strong need for public health educators to disseminate science-based information about ECs, including their potential health impacts and promising, but not yet proven efficacy for smoking cessation. We also found different motivational factors for EC use among the participants. Perceptions of EC use also varied by the users’ gender, age, and types of ECs they used. The Chinese social norm of sharing cigarettes among smokers may have promoted the dual use of ECs and traditional cigarettes, limiting the potential health benefits of switching to ECs. The present findings may inform future public health campaigns targeting the use, promotion, and regulatory measures of ECs in China and beyond.

## 7. Implications

This study adds to the literature about EC use perception among Chinese adults. Furthermore, our findings demonstrated the need to engage EC users in the development of EC use regulatory measures as their experience could provide useful insights about the potential use and perceptions of EC use in Chinese society. We found that the common reasons for using ECs were ease of use and carrying, hygiene, no fire hazard, aiding quitting smoking, reduced health hazards, etc. Most (90%; 27/30) participants reduced (77%) or quit smoking (13%) after using ECs, and 80% were willing to recommend these products to others. The Chinese social norm of sharing cigarettes among smokers may have promoted the dual use of ECs and traditional cigarettes, limiting the potential health benefits of switching to ECs. Future studies should pay attention to the associations between social norms and the use of traditional cigarettes and ECs.

## Figures and Tables

**Table 1 ijerph-17-02754-t001:** Demographic and cigarette use characteristics of study participants (*n* = 30).

Demographic Data	Counts (%)
Male	22 (73.3%)
Current use of traditional cigarettes	24 (80.0%)
Married	22 (73.3%)
Education level	
High school	3 (10.0%)
Undergraduate or graduate	27 (90.0%)
Employed	29 (96.7%)
**Cigarette Use Characteristics**	**Mean (SD)**
Age	35.5 (7.8)
Number of traditional cigarettes smoked per day	7.6 (8.3)
Years of smoking traditional cigarettes	13.0 (7.0)
Years since the first EC use	2.4 (2.0)

SD—standard deviation; EC—electronic cigarette.

**Table 2 ijerph-17-02754-t002:** Patterns and reasons for using ECs among the participants (*n* = 30).

EC Use Patterns	Counts (%)
First use EC through	
Self-purchase	10 (33.3%)
Introduced by others	20 (66.7%)
EC purchase channel	
Physical stores	13 (43.3%)
Online stores	13 (43.3%)
Bought by others	13 (43.3%)
Types of ECs used	24 (80.0%)
Heated liquid products	14 (46.7%)
Heated tobacco products	16 (53.3%)
Chargeable and replenishable	29 (96.7%)
Reasons for using ECs	
Curious about ECs	22 (73.3%)
To reduce cigarette smoking	25 (83.3%)
Addicted to EC	4 (13.3%)
Like the smell of the EC’s vapor	19 (63.3%)
Like the taste of the EC’s vapor	24 (80.0%)
Like the action and sensations of using ECs	11 (33.3%)
Can help me concentrate	5 (16.7%)
Can help me relax	17 (56.7%)
Refreshing and keeps me awake	16 (53.3%)
Cheaper than traditional cigarettes	8 (26.7%)
Know about nicotine concentration in ECs	17 (56.7%)
Use ECs every day	26 (86.7%)
Use ECs in non-smoking places	22 (73.3%)
Intend to quit ECs soon	0 (0%)
**EC Use Patterns**	**Mean (SD)**
EC cost per month (USD)	51.1 (32.6)
heated liquid product cost per month	40.3 (32.9)
heated tobacco product cost per month	61.2 (30.3)

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
