# Peer review of "Perception of E-cigarette Use among Adult Users in China: A Mixed-method Study"

_ijerph, 2020, doi:10.3390/ijerph17082754_

Round 1
Reviewer 1 Report
The manuscript presents results from an in-depth interview study that assessed the perceptions of EC use among adult users from a major metropolitan area in China. Their findings provide important information about the EC use pattern that are informative to the tobacco smoking prevention and control effort and regulatory policies related to EC. I have the following comments I think the authors should addressed in the revised version.
What percentage of people from the previous studies agreed to participate in the current study? Please give the response rate.
Was the 30 semi-structured in-depth interviews (IDI) pilot tested with EC user focus group? Please provide the IDI as supplemental information. Are there any conceptual theory (e.g. health belief, planned behavior theory, etc.) behind the five thematic sections of the interview questions?
Where any of the theme reached saturation? In other word, did the author found any saturation number for any of the themes identified?
While in the discussion age was mentioned as a potential factor influencing EC use and perception, the relevant results indicating this age differences were not presented in the Results section. Similarly, the discussion about health warning label was only mentioned in the discussion. Please provide the relevant results in the Results section as well.
Limitation should also mention that the participants were from Shanghai metropolitan area, which may not be generalizable to other regions or rural areas in China.
Are there any current survey about EC use in China available so that the authors can use the survey data to assess the representativeness of the study participants, in terms of user demographics, geographic residence, etc.?
Lines 130-131 on page 4 appears to be out of place.
Please be specific about “Most (90%; 27/30) participants have reduced (XX%?) or quit smoking (xx%?) after using ECs and 80% are willing to”
Author Response
Dear Reviewer,
We are grateful to you for your helpful comments. Following your and another reviewer’s suggestions, we have substantially improved the manuscript by making a number of changes. In cases where we have not made concrete changes, we also present below our reasoning behind our response. The changes in the text were marked in yellow highlight.
Reviewer 1
The manuscript presents results from an in-depth interview study that assessed the perceptions of EC use among adult users from a major metropolitan area in China. Their findings provide important information about the EC use pattern that are informative to the tobacco smoking prevention and control effort and regulatory policies related to EC. I have the following comments I think the authors should addressed in the revised version.
Comment 1: What percentage of people from the previous studies agreed to participate in the current study? Please give the response rate.
Response 1: 1037 participants across mainland China completed our online survey. At the end of that survey, we asked those who had been using e-cig for 6 months or longer and resided in the Shanghai and Kunshan (where Duke Kunshan University is located; approximately 45 miles from Shanghai) areas to contact us if they were interested in an interview. Thirty participants provided consent for participation in the interview. Thus the response rate is 2.9%. Please note that this rate would have been higher if there was no restriction on the locations of participants.
Comment 2: Was the 30 semi-structured in-depth interviews (IDI) pilot tested with EC user focus group? Please provide the IDI as supplemental information. Are there any conceptual theory (e.g. health belief, planned behavior theory, etc.) behind the five thematic sections of the interview questions?
Response 2: This was intended as a preliminary interview investigation of EC use and perception in users in mainland China. Although we sought feedback from several Chinese users during the process of developing the interview questions, we did not conduct a pilot study prior to this study.
Following your suggestion, we have now included the IDI as supplemental material.
As a preliminary investigation, the interview was focused on some major themes of EC use and perception that may be important for researchers, health professionals and the public (e.g., addictiveness, health impact, acceptance, etc.) rather than intended as a theory-based or thorough coverage of all aspects of the topic. We hope that current findings in the themes captured in this study could stimulate further theoretical and empirical research for better understanding of distinct social-cultural factors that affect EC use in China (e.g., the impact of cigarette sharing social norm, sex-specific motivations for EC use, etc).
Comment 3: Where any of the theme reached saturation? In other word, did the author found any saturation number for any of the themes identified?
Response 3: We did not follow data saturation. Our analyses were guided by the interview questions (i.e. a priori themes), which was developed based on the initial research aims, but emergent themes were also identified. Therefore, our approach combined an inductive and deductive approach.
Comment 4: While in the discussion age was mentioned as a potential factor influencing EC use and perception, the relevant results indicating this age differences were not presented in the Results section. Similarly, the discussion about health warning label was only mentioned in the discussion. Please provide the relevant results in the Results section as well.
Response 4: We are sorry for the omissions. We have added the relevant descriptions in the results section as follows.
“Because of the limited sample size and sampling method, we were unable to draw associations between age and e-cigarette use in this study. However, 9 participants mentioned that it was easier for youths to accept ECs as ECs were more fashionable and appealing to new smokers.”
“Most (79%) heated liquid EC users reported they rarely or never noticed health warning on the package. In contrast, this number is 50% for IQOS users, despite that warnings on IQOS were reported to be only printed in Japanese.”
Comment 5: Limitation should also mention that the participants were from Shanghai metropolitan area, which may not be generalizable to other regions or rural areas in China.
Response 5: Thank you for this suggestion. We have added this point to the discussion.
Comment 6: Are there any current survey about EC use in China available so that the authors can use the survey data to assess the representativeness of the study participants, in terms of user demographics, geographic residence, etc.?
Response 6: We thank the reviewer for this comment. We did not find any other similar studies in China. This should be noted that ours is a qualitative study among EC users in Shanghai area. Please also see the response #5 above.
Comment 7: Lines 130-131 on page 4 appears to be out of place.
Response 7: Thank you for the comment. We have adjusted its position.
Comment 8: Please be specific about “Most (90%; 27/30) participants have reduced (XX%?) or quit smoking (xx%?) after using ECs and 80% are willing to”
Response 8: We added the details: “Most (90%; 27/30) participants have reduced (77%) or quit smoking (13%) after using ECs and 80% are willing to recommend these products to others.”
Reviewer 2 Report
This study describes perceptions of and motivations for e-cigarette use among 30 Chinese adults who are current e-cigarette users.
This paper includes some interesting results, such as the high level of dual tobacco and EC use (80%) among participants, and participants’ describing the Chinese social norm of sharing cigarettes as a barrier to smoking cessation.
I have two major concerns about the paper that need to be addressed before the paper can be considered for publication.
First, the authors describe the study throughout as a qualitative study. However, as it contains quantitative results (e.g. Tables 1 and 2), it is in fact a mixed-methods study.
Second, I am concerned about the analytic approach, as the interview guide was already divided into five thematic sections (thus not allowing for the emergence of unexpected themes) and there is insufficient description of how analysis occurred, e.g. whether a deductive or inductive approach was used, and how the authors moved through the six steps of thematic analysis as outlined by Braun and Clarke. Much, much more detail is needed in this section. I am also confused as to how Excel can be adequately used for thematic analysis (I recommend NVivo instead). A clear description of exactly how the interview guide was created is needed. I worry that the authors undertook a very surface-level analysis, and this was limited by the interview guides’ questions, as the questions were already divided into themes, and close-ended questions are not a typical feature of qualitative studies. I am also concerned that interviews were not voice-recorded, and that all answers were hand-written; this means that some nuance may be missed, and some quotes may not be 100% correct. Also how was the sample size chosen- was thematic saturation reached?
A few other issues:
- I recommend a revision of the paper by a native English speaker or editing service, as there are multiple grammatical/spelling errors throughout the paper, e.g. “taking” nicotine on line 52, “quite” coffee shop on line 87, “with the recommendations from others” on line 123-124.
- I think the Introduction would benefit from a clearer description of similar studies conducted among Chinese participants, and a clearer description of the governance and regulation of ECs in China (if any)
- I recommend restructuring the results to ensure that presented finding/quotes actually fit within those themes. For example, under “Motivational Factors for using ECs”, the authors describe the participants’ preferred flavours (this is not a motivation for using ECs). Under “Satisfaction with ECs”, the authors describe on lines 197-198 that participants turn to traditional cigarettes when their craving for nicotine is strong- I think that this is better suited to the Motivation theme. These are only two examples of many that I found throughout the paper.
- Line 119- participants “currently smoked 7.6 cigarettes per day”- is this an average?
- Table 2 has formatting errors, and the difference between self-purchase and sales promotion is unclear, and unclear what RMB is or whether these figure are averages or medians?
- The paragraph of text (lines 132-136) below Table 2 is quantitative in nature (comparing differences between genders); this type of content does not belong in a qualitative paper
- Line 192- the quote is an example of a view of someone who thinks that ECs are more harmful- this doesn’t align with the text above. There is the same problem with the quote on lines 257-259- it contradicts the text above.
- Line 266- it is unclear what “stimulating” means in this context
- Line 274- Participants used ECs as a substitution- but 80% of participants are still smoking traditional cigarettes, so only 20% are using ECs as a substitution.
- For the finding about participants with inaccurate beliefs around the harms of nicotine, I recommend reviewing papers by Kylie Morphett and colleagues about vapers’ perceptions of the role of nicotine
- The ‘Implications’ section does not include any Implications. The paper is definitely lacking from a clear explanation of the utility and importance of these results.
Author Response
Dear Reviewer,
We are grateful to you for your helpful comments. Following your and other reviewers’ suggestions, we have substantially improved the manuscript by making a number of changes. In cases where we have not made concrete changes, we also present below our reasoning behind our response. The changes in the text were marked in yellow highlight.
Reviewer 2
This study describes perceptions of and motivations for e-cigarette use among 30 Chinese adults who are current e-cigarette users.
This paper includes some interesting results, such as the high level of dual tobacco and EC use (80%) among participants, and participants’ describing the Chinese social norm of sharing cigarettes as a barrier to smoking cessation.
I have two major concerns about the paper that need to be addressed before the paper can be considered for publication.
Comment 1: First, the authors describe the study throughout as a qualitative study. However, as it contains quantitative results (e.g. Tables 1 and 2), it is in fact a mixed-methods study.
Response 1: We agree that this study is a mixed-methods study since some of the interview questions asked for quantitative answers such as Likert scale ratings while others were open-ended. Nevertheless, considering the small sample size, we focused our analysis on pre-determined and emergent themes rather than quantitative differences. To avoid the confusion, we have removed “a qualitative study” from the title.
Comment 2: Second, I am concerned about the analytic approach, as the interview guide was already divided into five thematic sections (thus not allowing for the emergence of unexpected themes) and there is insufficient description of how analysis occurred, e.g. whether a deductive or inductive approach was used, and how the authors moved through the six steps of thematic analysis as outlined by Braun and Clarke. Much, much more detail is needed in this section. I am also confused as to how Excel can be adequately used for thematic analysis (I recommend NVivo instead). A clear description of exactly how the interview guide was created is needed. I worry that the authors undertook a very surface-level analysis, and this was limited by the interview guides’ questions, as the questions were already divided into themes, and close-ended questions are not a typical feature of qualitative studies. I am also concerned that interviews were not voice-recorded, and that all answers were hand-written; this means that some nuance may be missed, and some quotes may not be 100% correct. Also how was the sample size chosen- was thematic saturation reached?
Response 2: Thanks for the critique. We have now revised the analyses section (Section 2.3) to address your concerns.
Although we did not voice-record the interviews, two interviewers took notes during the conversation, which was reviewed immediately after each interview to ensure that inconsistencies are reduced as much as possible.
We choose the sample size based on the recommended sample size (sample size of 25) for qualitative studies. [Creswell JW. Qualitative inquiry and Research Design: Choosing among Five Approaches. 2006. California: Sage Publications.]
A few other issues:
1. I recommend a revision of the paper by a native English speaker or editing service, as there are multiple grammatical/spelling errors throughout the paper, e.g. “taking” nicotine on line 52, “quite” coffee shop on line 87, “with the recommendations from others” on line 123-124.
Response: We thank the reviewer for this reminder. We have revised this paper by asking help from a native English speaker.
2. I think the Introduction would benefit from a clearer description of similar studies conducted among Chinese participants, and a clearer description of the governance and regulation of ECs in China (if any).
Response: We tried to incorporate the limited data available about EC use in China. We have also elaborated briefly the governance and regulation of ECs in China.
3. I recommend restructuring the results to ensure that presented finding/quotes actually fit within those themes. For example, under “Motivational Factors for using ECs”, the authors describe the participants’ preferred flavours (this is not a motivation for using ECs). Under “Satisfaction with ECs”, the authors describe on lines 197-198 that participants turn to traditional cigarettes when their craving for nicotine is strong- I think that this is better suited to the Motivation theme. These are only two examples of many that I found throughout the paper.
Response: Thank you so much for this suggestion. We have adjusted the ones you mentioned and also incorporated several others.
4. Line 119- participants “currently smoked 7.6 cigarettes per day”- is this an average?
Response: Yes. We have added “on average.”
5. Table 2 has formatting errors, and the difference between self-purchase and sales promotion is unclear, and unclear what RMB is or whether these figure are averages or medians?
Response: Thank you. In this revised version, we have addressed these issues in Table 2.
6.The paragraph of text (lines 132-136) below Table 2 is quantitative in nature (comparing differences between genders); this type of content does not belong in a qualitative paper.
Response: We have changed this to “There were gender differences in the perceived patterns of EC use.”
7. Line 192- the quote is an example of a view of someone who thinks that ECs are more harmful- this doesn’t align with the text above. There is the same problem with the quote on lines 257-259- it contradicts the text above.
Response: The quote in line 192 is relevant to the previous paragraph. We have now made it clear by adding a statement as…. One EC user expressed this by saying:
“It is less harmful to other people. The vapor is not as strong as smoke from traditional cigarettes.” (Male, 22 years old, IQOS user).”
8. Line 266- it is unclear what “stimulating” means in this context
Response: We have changed the word to “strong”.
9. Line 274- Participants used ECs as a substitution- but 80% of participants are still smoking traditional cigarettes, so only 20% are using ECs as a substitution.
Response: This is correct. However, 77% of participants reduced their smoking of traditional cigarettes. We now have changed this sentence to “participants in this study reported that they used ECs as a substitution (as least partially) for traditional cigarettes and, participants believe that ECs helped them reduce smoking.”
10. For the finding about participants with inaccurate beliefs around the harms of nicotine, I recommend reviewing papers by Kylie Morphett and colleagues about vapers’ perceptions of the role of nicotine.
Response: Thank you for your suggestion. We cited this paper by adding “ECs users in Australia reported they also had difficulty sourcing practical and trustworthy information [35].”
11. The ‘Implications’ section does not include any Implications. The paper is definitely lacking from a clear explanation of the utility and importance of these results.
Response: We have now revised this section to include the implications.
Round 2
Reviewer 2 Report
The authors have substantially improved the paper; I commend them on their efforts.